# LaMPP: Language Models as Probabilistic Priors for Perception and Action

## Abstract

Language models trained on large text corpora encode rich distributional information about real-world environments and action sequences. This information plays a crucial role in current approaches to language processing tasks like question answering and instruction generation. We describe how to leverage language models for *non-linguistic* perception and control tasks. Our approach casts labeling and decision-making as inference in probabilistic graphical models in which language models parameterize prior distributions over labels, decisions and parameters, making it possible to integrate uncertain observations and incomplete background knowledge in a principled and query-efficient way. Applied to semantic segmentation, household navigation, and activity recognition tasks, this approach improves predictions on rare, out-of-distribution, and structurally novel inputs.[1]

## 1 Introduction

**Common-sense priors** are crucial for decision-making under uncertainty in real-world environments. Suppose that we wish to label the objects in the scene depicted in Fig. 1(b). Once a few prominent objects (like the bathtub) have been identified, it is clear that the picture depicts a bathroom. This helps resolve some more challenging object labels: the curtain in the scene is a shower curtain, not a window curtain; the object on the wall is more likely a mirror than a painting. Prior knowledge about likely object or event co-occurrences are essential not just in vision tasks, but also for navigating unfamiliar places and understanding other agents' behaviors. Indeed, such expectations play a key role in human reasoning for tasks like object classification and written text interpretation (Kveraga et al., 2007; Mirault et al., 2018).

In most problem domains, current machine learning models acquire information about the prior distribution of labels and decisions from task-specific datasets. Especially when training data is sparse or biased, this can result in systematic errors, particularly on unusual or out-of-distribution inputs. How might we endow models with more general and flexible prior knowledge?

We propose to use **language models (LMs)**—learned distributions over natural language strings—as task-general probabilistic priors. Unlike segmented images or robot demonstrations, large text corpora are readily available and describe diverse facets of human experience. LMs trained on them encode much of this information—like the fact that *plates are located in kitchens*, and that *whisking eggs is preceded by breaking them*—with greater diversity and fidelity than provided by most task-specific datasets. Such linguistic supervision has also been hypothesized to play a role in aspects of human common-sense knowledge that are difficult to learn from direct experience (Painter, 2005).

In text generation tasks, LMs are used as knowledge sources in tasks spanning common-sense question answering (Talmor et al., 2021), storytelling (Ammanabrolu et al., 2020; 2021), and program synthesis (Lew et al., 2020). They have also been applied to grounded language understanding problems via "model chaining" approaches, which encode the output of perceptual systems as natural language strings that can be input to LMs (Zeng et al., 2023; Singh et al., 2022). In general, such approaches are effective but computationally costly to run—LMs must be queried for *every possible labeling decision*, limiting their applicability to problems with simple action or label spaces.

---

[1] Our code will be made publicly available upon acceptance.

In this paper, we focus on LMs as a source of probabilistic background knowledge that can be integrated with existing domain models and inference procedures. By using LMs to place prior distributions over labels, decisions or model parameters, we can combine "top-down" background knowledge with "bottom-up" task models, resulting in a principled framework for integrating linguistic supervision with structured uncertainty about non-linguistic variables.

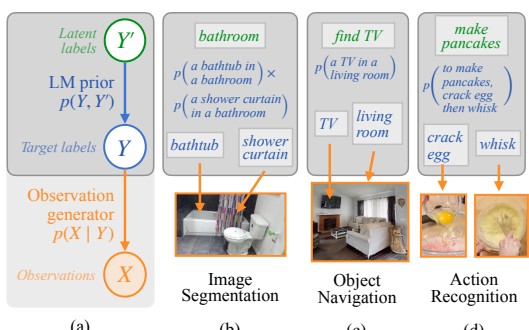

We call this approach **LAMPP** (**La**nguage **M**odels as **P**robabilistic **P**riors). LAMPP is a generic modeling framework that operates on top of and improves base models for existing tasks. Similar in spirit to chain-of-thought prompting (Wei et al., 2022) and Socratic model approaches (Zeng et al., 2023), LAMPP is not a (single) model that can be applied in the same way across domains, but rather a "modeling philosophy" based on querying LMs for the *parameters of task-specific priors* rather than for predictions directly. Our experiments show that this approach can match or outperform existing "prediction-first" approaches for integrating LMs and domain models, while requiring orders of magnitude fewer LM queries.

Figure 1: In LAMPP, the LM provides a prior over a structured label space $P(Y, Y')$ and a task-specific observation model provides $P(X \mid Y)$. We apply LAMPP to concrete tasks, including image segmentation and video action recognition. In image segmentation, the LM provides a prior over what objects are likely to co-occur (based on room-object probabilities), which allows it to determine that the observed curtain is a *shower curtain*. In action recognition, the LM provides a prior over what action sequences are likely to accomplish the target tasks, allowing it to infer the action sequence in a video.

We present three case studies featuring tasks with diverse objectives and input modalities—semantic image segmentation, robot navigation, and video action recognition. Across problem domains, LAMPP consistently improves performance on rare, out-of-distribution, and structurally novel inputs, and sometimes in-distribution accuracy. We also show that LAMPP offers *complementary* benefits to multimodal pre-training schemes (like CLIP (Radford et al., 2021)) that learn from paired text and task data. Our results show that language is a useful source of background knowledge for general decision-making, and that uncertain background knowledge can be integrated with uncertain observations to yield accurate predictions. Moreover, LaMPP interacts with LMs once per domain rather than once per input, it can be applied to large test sets and label spaces with no query overhead.

## 2 METHOD

A language model (LM) is a distribution over natural language strings. LMs trained on sufficiently large text datasets become good models not just of grammatical phenomena, but various kinds of world knowledge (Talmor et al., 2021; Li et al., 2021). Our work proposes to extract probabilistic common-sense priors from language models, which can then be used to supplement and inform *arbitrary* task-specific models operating over multiple modalities. These priors can be leveraged at multiple stages in the machine learning pipeline:

**Prediction:** In many learning problems, our ultimate goal is to model a distribution $p(y \mid x)$ over labels or decisions $y$ given (non-linguistic) observations $x$. These $y$s might be structured objects: in Fig. 1(b), $x$ is an image and $y$ is a set of labels for objects in the image. By Bayes' rule, we can write $p(y \mid x) \propto p(y)p(x \mid y)$, which factors this decision-making problem into two parts: a prior over labels $p(y)$, and a generative model of observations $p(x \mid y)$. With such a generative model, we may immediately combine it with a representation of the prior $p(y)$ to model the distribution over labels.

**Learning:** In models with interpretable parameters, we may also leverage knowledge about the distribution of these parameters themselves during learning, before we make any predictions at all. Given a dataset $\mathcal{D}$ of examples $(x_i, y_i)$ and a predictive model $p(y \mid x; \theta)$, we may write:

$$p(\theta \mid \mathcal{D}) \propto p(\mathcal{D} \mid \theta)p(\theta) = \Big( \prod_i p(y_i \mid x_i; \theta) \Big) p(\theta) \,,$$

in this case making it possible to leverage prior knowledge of $\theta$ itself, e.g., when optimizing model parameters or performing full Bayesian inference.

In structured output spaces like segmented images or robot trajectories, a useful prior contains information about which joint configurations are plausible (e.g., an image might contain sofas and chairs, or showers and sinks, but not sinks and sofas). How can we use an LM to obtain and use distributions $p(y)$ or $p(\theta)$? Applying LAMPP in a given problem domain involves four steps:

1. **Choosing a base model**: We use any model of observations $p(x \mid y)$ or labels $p(y \mid x; \theta)$.

2. **Designing a label space**: When reasoning about a joint distribution over labels or parameters, correlations between these variables might be expressed most compactly in terms of some other latent variable (in Fig. 1(b), object labels are coupled by a latent *room*). Before querying an LM to obtain $p(y)$ or $p(\theta)$, we may introduce additional variables like this one to better model probabilistic relationships among labels.

3. **Querying the LM**: We then obtain scores for each configuration of $y$ or $\theta$ by *prompting* a language model with a query about the plausibility of the configuration, then *evaluating* the probability that the LM assigns to the query. Examples are shown in Fig. 1(b–c). For all experiments in this paper, we use the GPT-3 (specifically, the `text-davinci-003` model) to score queries (Brown et al., 2020).

4. **Inference**: Finally, we perform inference in the graphical model defined by $p(y)$ and $p(x \mid y)$ (or $p(\theta)\, p(y \mid x, \theta)$) to find the highest-scoring configuration of $y$ for a given $x$.

In Sections 3–5, we apply this framework to three learning problems. In each section, we evaluate LAMPP's ability to improve *generalization* over base models. We focus on three types of generalization: **zero-shot** (ZS), **out-of-distribution** (OOD), and **in-distribution** (ID). The type of generalization required depends on the availability and distribution of training data: ZS evaluations focus on the case in which $p(x \mid y)$ is known (possibly just for components of $y$, e.g., appearances of individual objects), but no information about the joint distribution $p(y)$ (e.g., configurations of rooms) is available at training time. OOD evaluations focus on biased training sets (in which particular label combinations are over- or under-represented). ID evaluations focus on cases where the full evaluation distribution is known and available at training time.

Finally, we note that LAMPP is much cheaper than approaches in which the LM is responsible for directly generating final predictions. In LAMPP, there is a fixed overhead cost to construct the graphical model for each domain, after which inference does not require interacting with an LM at all; model probabilities can be re-composed and reused indefinitely for each new inference.

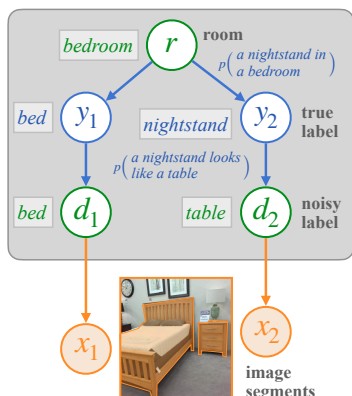

Figure 2: Generative model for **image semantic segmentation**. Images originate in a room $r$, which generates the objects $y_1, y_2$ in the room, which generate noisy labels $d_1, d_2$ representing perceptually similar objects. Finally, each $d_i$ generates an image segment $x_i$, a continuous region of image pixels depicting each object. Rooms $r$, true labels $y_i$, and noisy labels $d_i$ are latent, while image segments $x_i$ are observed.

## 3 LAMPP FOR SEMANTIC SEGMENTATION

We first study the task of **semantic image segmentation**: identifying object boundaries in an image and labeling each object $x_i$ with its class $y_i$. How might background knowledge from an LM help with this task? Intuitively, it may be hard for a bottom-up visual classifier to integrate global image context and model correlations among distant objects' labels. LMs encode common-sense information about the global structure of scenes, which can be combined with easy-to-predict object labels to help with more challenging predictions.

### 3.1 METHODS

**Base model** Standard models for semantic segmentation discriminatively assign a label $y_i$ to each pixel $x_i$ in an input image $x$ according to some $p_{\text{seg}}(y_i \mid x)$. We use RedNet (Jiang et al., 2018), a ResNet-50-based autoencoder model, or CLIPSeg (Lüddecke & Ecker, 2022), a zero-shot segmentation model built off CLIP,

| Base Model + Method | Total # Tokens ($ Cost) | Best / Worst / Avg. $\Delta$IoU | $\Delta$ IoU over object classes |
|---|---|---|---|
| RedNet (ID) + LAMPP | 9.5e+5 tokens ($1.86*) | +18.9 / −2.16 / +0.5 | |
| RedNet (ID) + SM | 1.6e+8 tokens ($3112*) | +16.9 / −37.2 / −10.3 | |
| RedNet (OOD) + LAMPP | 9.5e+5 tokens ($1.86*) | +8.92 / −2.50 / +0.2 | |
| CLIPSeg (ZS) + LAMPP | 9.5e+5 tokens ($1.86*) | +13.3 / −5.0 / +0.4 | |

Table 1: Image semantic segmentation results for ID and OOD generalization. We report the total cost (in terms of # tokens, and the rough dollar amount) for running the entire experiment. LAMPP is much cheaper than SM. We also report the improvement to Intersection-over-Union (IoU) when applying either LAMPP or a Socratic model approach to each base model. We compute separate $\Delta$IoU for each object class and report the largest, smallest, and average $\Delta$IoU over classes in the third column, visualizing the full distribution of $\Delta$IoU over object classes in the rightmost column. LAMPP improves semantic segmentation dramatically on certain categories, while having minimal effect on all other categories. *Costs computed relative to Sept. 28, 2023 pricing.

to compute $p_{\text{seg}}$. By computing $\arg\max_y p(y \mid x)$ for each pixel in an input image, we obtain a collection of **segments**: contiguous input regions assigned the same label (see Fig. 2 bottom). When applying LAMPP, we treat these segments as given, but infer a new joint labeling.

**Label Space** We hypothesize a generative process (Fig. 2) in which every image originates in a **room** $r$. Conditioned on the room, a fixed number of **objects** are generated, each with label $y_i$. To model possible perceptual ambiguity, each true object labels in turn generates a **noisy** object label $d_i$. Finally, each of these generates an image segment $x_i$.

We use the base segmentation model $p_{\text{seg}}$ to compute $p(x_i \mid d_i)$ by applying Bayes' rule locally for each segment: $p(x_i \mid d_i) \propto p_{\text{seg}}(d_i \mid x_i)/p(d_i)$. All other distributions are parameterized by a LM. Ultimately, we wish to recover object labels $y_i$; latent labels $r$ and $d$ help extract usable background information about objects' co-occurrence patterns and perceptual properties.

**LM Queries** We compute the object–room co-occurrence probabilities $p(y_i \mid r)$ by prompting the LM with: "*A(n) [r] has a(n) [$y_i$]*: [*plausible / implausible*] ". We compute the relative probability assigned to the tokens *plausible* and *implausible*, then normalize over all object labels $y$ to parameterize the final distribution. We use the same procedure to parameterize the object–object confusion model $p(d_i \mid y_i)$, using the prompt: "*The [$d_i$] looks like the [$y_i$]*: [*plausible / implausible*] ".

**Inference** The model in Fig. 2 defines a joint distribution over all labels $\underline{y} = y_1, \ldots, y_n$. To re-label a segmented image, we compute the max-marginal-probability label for each segment independently:

$$\arg\max p(y_i \mid \underline{x}) = \arg\max \sum_r \sum_{\underline{y} \setminus \{y_i\}} \sum_{\underline{d}} p(\underline{x}, \underline{d}, \underline{y}, r) \tag{1}$$

The form of the decision rule used for semantic segmentation (which includes several simplifications for computational efficiency) can be found in Appendix A.1.

## 3.2 EXPERIMENTS

We use the SUN RGB-D dataset (Song et al., 2015), which contains RGB-D images of indoor environments. We also implement a Socratic model (SM) baseline that integrates LM knowledge without considering model uncertainties. We take noisy labels from the image model ($d_i$) and directly

query the LM for true labels ($y_i$). Details of this baseline can be found in Appendix A.2. We evaluate the RedNet and CLIPSeg *base models*, this *Socratic model* approach, and LAMPP on **in-distribution**, **out-of-distribution**, and **zero-shot** generalization.

**ID Generalization**   We use a RedNet checkpoint trained on the entire SUNRGB-D training split. As these splits were not created with any special biases in mind, the training split should reflect a similar label distribution to the test split.

**OOD Generalization**   We train RedNet on a distribution $p(y_i, y_j)$ that differs from the true distribution. We do this by picking two object labels that commonly occur together (i.e. picking $y_i$ and $y_j$ such that $p(y_i, y_j)$ is high), and removing all images from the training set where they *do* occur together (thus making $p(y_i, y_j)$ close to zero in the training set). In this case, we choose *beds* and *nightstands* and remove images where they co-occur from the training set. We evaluate on the original test split where beds and nightstands frequently co-occur.

**ZS Generalization**   We use CLIPSeg applied to the SUNRGB-D domain without fine-tuning. We use CLIPSeg instead of RedNet here because CLIPSeg is pre-trained and designed to be applied zero-shot, whereas RedNet is not pre-trained.

### 3.3 RESULTS

We measure change in intersection-over-union ($\Delta$IoU) between predicted and ground-truth object segmentations afforded to each object class after applying LAMPP and SM to each base model. Results can be found in Table 1 and Appendix A.3.

We see that in all cases, LAMPP improves the accuracy of the base classifier. To get a better understanding of the distribution of improvements over object categories, we report *per-category differences in IoU* of our model *relative* to the baseline image model (columns 2 and 4). As expected LAMPP offers very large accuracy improvements on a small number of rare classes, while preserving model behavior on all other classes. The third column shows that LAMPP is much cheaper than SM. Note that we do not run SM in the OOD or ZS cases due to its prohibitive cost.

In the **ID setting**, the accuracy of detecting *shower curtains* (see Appendix A.3) improves by nearly 20 points with LAMPP, as the base model obtains near-0% mIoU on shower curtains, almost always mistaking them for *(window) curtains*. Here, background knowledge from language fixes a major (and previously undescribed) prediction error for a rare class. The Socratic model approach repairs prediction errors on the same rare class as LAMPP in the ID setting, but introduces many other errors.

In the **OOD setting**, the base image model sees far fewer examples of nightstands and consequently never predicts nightstands on the test data. (*Nightstands* are frequently predicted to be *tables* and *cabinets* instead). This is likewise rectified with LAMPP: background knowledge from language reduces model sensitivity to a systematic bias in dataset construction.

Finally, in the **ZS setting**, we see that LAMPP is able to offer additional improvements even on top of a model trained in a multi-modal fashion. This shows that even large, pre-trained multi-modal models are subject to biases that can be fixed with (other) LMs.

## 4 LAMPP FOR NAVIGATION

We next turn to the problem of **object navigation**. Here, we wish to build an agent that, given a goal object $g$ (e.g., a television or a bed), can take actions $a$ to explore and find $g$ in an environment, while using noisy partial observations $x$ from a camera for object recognition and decision-making. Prior knowledge about where goal objects are likely located can guide this exploration, steering agents away from regions of the environment unlikely to accomplish the agent's goals.

### 4.1 METHODS

**Base model**   We assume access to a pre-trained navigation policy (in this case, from the STUBBORN agent; Luo et al., 2022) that can plan a path to any specified coordinate $a$ in the environment given image observations $x$. Our goal is to build a *high-level* policy $\pi(a \mid x)$ that can direct this low-level navigation. We focus on household environments, and assume access to a coarse semantic map of

an environment that identifies rooms, but not locations of objects within them. In each state, the STUBBORN low-level navigation policy also outputs a scalar score reflecting its confidence that the goal object is present.

**Label space**  Our high-level policy alternates between performing two kinds of actions $a$: **Navigation**, where the agent chooses a room $r$ in the environment to move to (when a room is selected, we direct the low-level navigation policy to move to a point in the center of the room, and then explore randomly within the room for a fixed number of time steps), and **Selection**, whenever an observation is received *during* navigation, the agent evaluates whether it the goal object is present (if so, the episode ends).

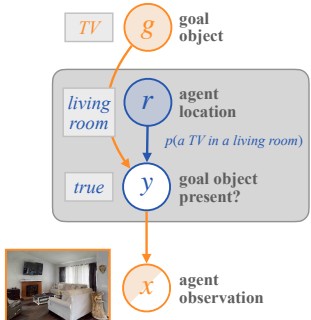

Figure 3: Generative model for **object navigation**. Given goal object $g$, we depict a decomposition of the agent's *success score* $y$, which takes on value 1 (*true*) if $g$ is present and 0 (*false*) otherwise. We focus on a household domain in which agents navigate to rooms $r$. $r$ generates the success condition $y$ (indicating whether $g$ is present at the agent location), which generates the agent observation $x$ of the location. Goal objects $g$ and rooms $r$ are given, success conditions $y$ are latent, and agent observations $x$ are partially observed.

A rollout of this policy thus consists of a sequence of navigation actions, interleaved with a selection action for every observation obtained while navigating. In both cases, choosing actions effectively requires inference of a specific unobserved property of environment state: whether the goal object is in fact present near the agent. We represent this property with a latent variable $y$. When navigating, the agent must infer the room that is most likely to contain the goal object. When selecting, the agent must infer whether its current perception is reliable.

We normalize the low-level policy's success score and interpret it as a distribution $p(x \mid y)$, then use the LM to define a distribution $p(y \mid r, g)$. Together, these give a distribution over latent success conditions and observations given goals and locations, which are used to select high-level actions.

**LM queries**  For $p(y \mid r, g)$, we use the same query as in Section 3 for deriving object–room probabilities, inserting $g$ in place of $y_i$, except here we do not normalize over object labels (since $y$ is binary), and simply take the relative probability of generating the token *plausible*.

**Inference**  With this model, we define a policy that performs inference about the location of the goal object, then greedily attempts to navigate to the location most likely to contain it. This requires defining $p(a \mid x, g)$ for both navigation and selection steps. In **Navigation**, the agent chooses a room $r$ maximizing $p(y \mid r, g)$. (The agent does not yet have an observation from the new room, so the optimal policy moves to the room most likely to contain the goal object *a priori*.) In **Selection**, the agent ends the episode only if $p(y \mid x, r, g) > \tau$ for some confidence threshold $\tau$. We use $\tau = 0.2$ in our experiments, tuned on a subset of the training data.

During exploration, the agent maintains a list of previously visited rooms. Navigation steps choose only among rooms that have not yet been visited.

## 4.2  EXPERIMENTS

We consider a modified version of the Habitat Challenge ObjectNav task (Yadav et al., 2022). The task objective is to find and move to an instance of the object in unfamiliar household environments as quickly as possible. The agent receives first-person RGBD images, compass readings, and 2D GPS values as inputs at each timestep. In our version of the task, we assume access to a high-level map of the environment which specifies the coordinates and label of each room. Individual objects are not labeled; the agent must rely on top-down knowledge of where certain objects are likely to be in order to efficiently find the target object.

We implement a SM baseline where the LM guides agent exploration by specifying an ordering of rooms to visit. Details can be found in Appendix B.1. This is similar to prior work that use LMs to specify high-level policies (Zeng et al., 2023; Sharma et al., 2022), whereby neither LM nor observation model uncertainties are accounted for when generating the *high-level* policy.

We modify the base STUBBORN agent to condition on the high-level map. The base model visits rooms in a random order (effectively beginning with a uniform belief over object locations given

rooms). Results in Appendix B.4 show that an agent equipped with this high-level policy performs comparably to the original STUBBORN agent. We evaluate how much *Socratic modeling* or LAMPP improves the ability of this *Uniform Prior* agent to perform **zero-shot generalization**, where the training data does not contain any information about $p(y \mid r, g)$.[2]

## 4.3 EVALUATION & RESULTS

We evaluate success rate (SR), the percent of instances in which the agent successfully navigated to the goal object.[3] Because the STUBBORN agent is designed to handle only single floors we evaluate only instances in which the goal object is located on the same floor as the agent's starting location.

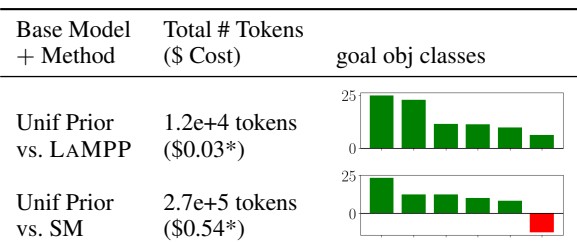

| Base Model + Method | Total # Tokens ($ Cost) | goal obj classes |
|---|---|---|
| Unif Prior vs. LAMPP | 1.2e+4 tokens ($0.03*) | |
| Unif Prior vs. SM | 2.7e+5 tokens ($0.54*) | |

Table 2: Navigation Results for ZS generalization. We report improvements to success rates (SR) when applying either LAMPP or SM to a base uniform priors model. We compute separate ΔSR for each goal object class and report the largest, smallest, and average ΔSR over goal objects in the second column. We visualize the full distribution of ΔSR over goal objects in the rightmost column. By using LAMPP, we are able to achieve significant improvement over almost all object classes. *Costs computed relative to Sept. 28, 2023 pricing.

Results are reported in Table 2 and Appendix B.3. We report ΔSR on each goal object from applying either LAMPP or SM to the base model. Using LAMPP, we find greatest improvements in goal object categories that have strong tendencies to occur only in specific rooms, such as *TV monitors* (+33.3), and smallest for objects that occur in many different rooms, like *plants* (+0.0) (Appendix B.3). Compared to SM, LAMPP improves SR by a wide margin (see Appendix B.5 for analysis). LAMPP is also substantially more query-efficient: SM requires one query *per navigation action*, while LAMPP performs a fixed number of initial queries, which are reused across all actions and episodes.

## 5 LAMPP FOR ACTION RECOGNITION AND SEGMENTATION

The final task we study focuses on video understanding: specifically, taking demonstrative videos of a task (e.g., making an omelet) and segmenting them into actions (e.g., cracking or whisking eggs). Because it is hard to procure segmented and annotated videos, datasets for this task are usually small, and it may be difficult for models trained on task data alone to learn robust models of task-action relationships and action orderings. Large LMs' training data contains much more high-level information about tasks and steps that can be taken to complete them.

## 5.1 METHODS

**Base model** Given a video of task $t$, we wish to label each video frame $x_i$ with an action $y_i$ (chosen from a fixed inventory of plausible actions for the task) according to $\arg\max_{y_1 \cdots y_n} p(y_1 \cdots y_n \mid x_1 \cdots x_n, t)$. We build on a model from Fried et al. (2020) that frames this as inference in a task-specific hidden Markov model (HMM) in which a latent sequence of actions generates a sequence of video frames according to:

$$p(x_{1 \cdots n} \mid y_{1 \cdots n}) \propto \prod_j p(x_j \mid y_j; \eta) \, p(y_j \mid y_{j-1}; \theta) \qquad (2)$$

(omitting the dependence on the task $t$ for clarity). This generative model decomposes into an **emission model** with parameters $\eta$ and a **transition model** with parameters $\theta_t$, and allows efficient inference of $p(y \mid x)$.[4] $p(y_j \mid y_{j-1}; \theta)$ is a multinomial distribution parameterized by a transition matrix encoding probability each action $y_{j-1}$ is followed by action $y_j$.

---

[2]At the time these experiments were conducted, room labels were not yet present in the dataset, so we could only study the zero-shot setting. To evaluate LAMPP, the first two authors of the paper manually annotated room labels in the evaluation set.

[3]We report Success weighted by Path Length (SPL) in Appendix B.3.

[4]The model in Fried et al. (2020) is a hidden *semi*-Markov model (HSMM) in which latent action states generate multiple lower-level actions in sequence. We omit the HSMM emission model for clarity of presentation.

Here we apply LaMPP to the problem of learning model parameters themselves. We use a LM to place a *prior on transition parameters* $\theta$, thus incorporating prior knowledge from LMs while learning action transition distributions from data. Given a dataset of labeled videos of the form $(x_{1...n}, y_{1...n})$, we compute a maximum *a posteriori* estimate of $\theta$:

$$\arg\max_\theta \ \log p(\theta) + \sum_{x,y} \sum_j \log p(y_j \mid y_{j-1}; \theta) , \qquad (3)$$

(likewise for $\eta$).

**Label space** We parameterize the prior $p(\theta)$ as a Dirichlet distribution with hyperparameters $\alpha$, according to which $p(\theta) \propto \prod_i \theta_i^{\alpha_i - 1}$. Intuitively, the larger $\alpha_i$ is, the more probable the corresponding $\theta_i$ is judged to be *a priori*. Here, parameters $\theta_{y \to y'}$ are probabilities of transitioning from action $y$ to $y'$; we would like $\alpha_{y \to y'}$ to be large for plausible transitions.

**Prompting the LM** To derive values of $\alpha$ for each action transition $y \to y'$, we query the LM with the prompt: "*Your task is to* [$t$]. *Here is an \*unordered\* set of possible actions: {[Y]}. Please order these actions for your task. The step after* [$y$] *can be* [$y'$]", where $Y$ is a set of all available actions for the task. We condition the LM on the non-highlighted portion of the prompt and set $\alpha_{y \to y'} = \lambda \cdot p_{\text{LM}}(y' \mid \text{prompt}(y))$, where $\lambda$ controls the strength of the prior.

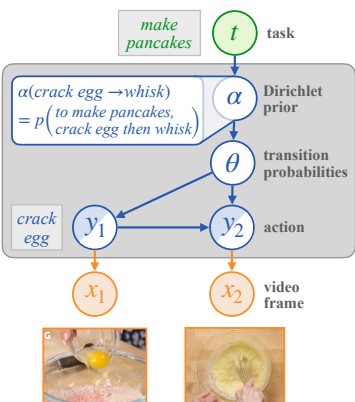

Figure 4: Generative model for **action segmentation**. The base model is a HMM with transition probabilities parameterized by $\theta$. We generate a prior over *model parameters* $\theta$: Each task $t$ generates a Dirichlet prior $\alpha$ over action transitions, which in turn generates $\theta$. $\theta$ parameterizes the action transition distribution $y_1 \to y_2$. Each action $y_i$ at timestep $i$ then generates the observed video frame $x_i$. Tasks $t$ and video frames $x_i$ are observed, actions $y_i$ are partially-observed, and parameter priors $\alpha$ and parameters $\theta$ are latent.

**Inference** The use of a Dirichlet prior means that Eq. (3) has a convenient closed-form solution:

$$\theta_{y \to y'} = \frac{\alpha_{y \to y'} + \#(y \to y') - 1}{\left(\sum_{y''} \alpha_{y \to y''}\right) + \#(y) - |Y|} , \qquad (4)$$

where $\#(y \to y')$ denote number of occurrences of the transition $y \to y'$ in the training data, $\#(y)$ denotes number of occurrences of $y$ in the training data, and $|Y|$ is total number of actions.

## 5.2 EXPERIMENTS

We use the CrossTask dataset (Zhukov et al., 2019), which features instructional videos depicting tasks (e.g., *make pancakes*). The learning problem is to segment videos into regions and annotate each region with the corresponding action being depicted (e.g., *add egg*). We evaluate the ability of the LaMPP to improve the **zero-shot** and **out-of-distribution** generalization abilities of the base model. For all experiments with LaMPP, we use $\lambda = 10$. We do not study an SM baseline for this task, as they are unable to generate parameters rather than labels.[5]

**ZS Generalization** We assume that the training data contains no information about the transition distribution $p(y_i \mid y_{i-1}, t)$. However, we still assume access to *all video scenes and their action labels*, which allows us to learn emission distributions $p(x_i \mid y_i)$. We do this by assuming access to only an *unordered set* of video frames from each task, where each frame is annotated with its action label, but with no sense of which frame preceded or followed it.

Because we have no access to empirical counts of transitions from the training data, the model falls back completely on its priors when computing those parameters $\theta_{y \to y'} = \frac{\alpha_{y \to y'} - 1}{\left(\sum_{y''} \alpha_{y \to y''}\right) - |Y|}$ which is uniform for the base model and derived from the LM for LaMPP.

**OOD Generalization** We bias the *transition distribution* by randomly sampling a common transition from each task and holding out all videos from the training set that contain that transition.

---

[5]We also find that LaMPP did not improve the base model in-domain. We believe this is because the base model is simple enough that simply counting transitions and observation co-occurrence is sufficient for estimating parameters, and the action space is small enough that the dataset sufficiently covers the actions and transitions for each task.

## 5.3 Evaluation & Results

Following Fried et al. (2020), we evaluate *step recall*, i.e. the percentage of actions in the real action sequence that are also in the model-predicted action sequence. For simplicity, we ignore background actions during evaluation. Results are shown in Table 3 and Appendix C.2. For both the ZS and OOD settings, step recall slightly improves with LAMPP. The small magnitude of improvement may be because the LM sometimes does not possess a sensible prior over action sequences: for example, it is biased towards returning actions in the order they are named in the prompt.

| Base Model + Method | # Tokens ($ Cost) | Δ Recall (class avg. / freq avg.) |
|---|---|---|
| HMM (ZS) + LAMPP | 5.4e+5 ($1.09*) | +1.3 / +1.9 |
| HMM (OOD) + LAMPP | 5.4e+5 ($1.09*) | +0.5 / +0.3 |

Table 3: Video segmentation results for ZS and OOD generalization. We report the average improvement in step recall for LAMPP applied to the base HMM model from Fried et al. (2020). We report both a *class-averaged* step recall (over actions) and a *frequency-averaged* step recall (over videos). We also report the distribution of Δ recall over *tasks* in the rightmost column. Note in the ZS case, LAMPP provides significant improvement in certain task classes. *Costs relative to Sept. 28, 2023 pricing.

## 6 Related Work

**String Space Model Chaining**   There has been much recent work in combining and composing the functionality of various models *entirely in string space*. The Socratic models framework (Zeng et al., 2023) proposes chaining together models operating over different modalities by converting outputs from each into natural language strings. Inter-model interactions are performed purely in natural language. While such methods have yielded strong results in tasks like egocentric perception and robot manipulation (Ahn et al., 2022), they are fundamentally limited by the expressivity of the string-valued interface. Models often output useful features that cannot be easily expressed in language, such as graded or probabilistic uncertainty (e.g., in a traditional image classifier). Even if such information is written in string form, there is no guarantee that language models will correctly use it for formal symbolic reasoning—today's LLMs still struggle with arithmetic tasks expressed as string-valued prompts. (Ye & Durrett, 2022). Thus, string-space approaches are limited when outputs from task-specific models involve gradation or uncertainty that is not easily expressed in language.

Concurrent to the present work is LMPriors (Choi et al., 2022), which similarly seeks to use language model scores as a source of common-sense information in other decision-making tasks. There, LMs are applied to feature selection, reward shaping, and casual inference tasks, rather than explicit probabilistic models. However, the approach involves sampling greedily from the prior distribution of the language model over variable names, discarding any/all graded information about the relationship between variables. We therefore explore how language model outputs can be worked with probabilistically instead.

**LMs and Probabilistic Graphical Models**   Many prompting methods for language processing tasks, like chain-of-thought prompting (Wei et al., 2022) and bootstrapped rationale-generation (Zelikman et al., 2022) may be interpreted as probabilistic programs built from LM queries (Dohan et al., 2022). However, this analysis exclusively considers language tasks; to the best of our knowledge, the present work is the first to specifically connect language model evaluations to probabilistic graphical models in non-language domains.

## 7 Conclusion

We present LAMPP, a generic technique for integrating background knowledge from language into decision-making problems by extracting probabilistic *priors* from language models. LAMPP improves zero-shot, out-of-distribution, and in-distribution generalization across image segmentation, household navigation, and video-action recognition tasks. It enables principled composition of uncertain perception and noisy common-sense and domain priors, and shows that language models' comparatively unstructured knowledge can be integrated naturally into structured probabilistic approaches for learning or inference. The effectiveness of LAMPP depends crucially on the quality of the LMs used to generate priors. While remarkably effective, today's LMs still struggle to produce calibrated plausibility judgments for some rare tasks. As the quality of LMs for core NLP tasks improves, we expect that their usefulness for LAMPP will improve as well.

ETHICS STATEMENT

Large language models are known to learn false or misleading information and socially-biased distributions from their pre-training data. Blind application of these priors to various domains can thus result in potentially erroneous or biased predictions. As seen above, improvements at assigning specific labels tasks are tied to the accuracy of the priors over those labels. An important benefit of LAMPP is that these priors have simple, explicit representations (as conditional probability tables rather than large neural models). However, even these will inherit biases and errors from the LMs that generate them, and human review is essential before deploying LAMPP in real-world scenarios.

REPRODUCIBILITY

We will make the code publicly available upon acceptance. All prompts we used for querying GPT* models are available in the Appendix. In our experiments, we queried the GPT3 `text-davinci-002` model for token probabilities over output tokens. Further details on hyperparameter settings and computational requirements can be found in Appendices D and E.

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

# A LaMPP for Semantic Segmentation

## A.1 Methods

We derive the following decision rule from the model in Fig. 2:

$$p(y_i \mid \underline{x}) \overset{\propto}{\sim}$$

$$p(y_i \mid d_i = d_i^*)p(d_i = d_i^* \mid x_i)\left(\sum_r p(r)p(y_i \mid r)\prod_{j=1\cdots n}\left(\sum_{y_j}\frac{p(r \mid y_j)p(d_j = y_j \mid x_j)}{p(r)}\right)\right) \tag{5}$$

We obtain this decision rule as described below. Here we denote rooms $r$, true object labels $y$, noisy object labels $d$, and observations $x$. (Underlines denote sets of variables, so e.g., $\underline{x} = \{x_1, \ldots, x_n\}$.) Finally, we write $d_i^*$ to denote the base model's prediction for each image segment ($d_i^* = \arg\max p_{\text{seg}}(d_i \mid x_i)$). To see this:

$$p(y_i \mid \underline{x}) \propto \sum_r \sum_{\underline{y}\backslash\{y_i\}} \sum_{\underline{d}} p(\underline{x}, \underline{y}, \underline{d}, r)$$

$$= \sum_r \sum_{\underline{y}\backslash\{y_i\}} \sum_{\underline{d}} p(r)\Big(p(y_i \mid r)p(d_i \mid y_i)p(x_i \mid d_i)\Big)\prod_j p(y_j \mid r)p(d_j \mid y_j)p(x_j \mid d_j)$$

$$= \sum_r p(r)\Big(p(y_i \mid r)\sum_{d_i} p(d_i \mid y_i)p(x_i \mid d_i)\Big)\Big(\prod_j \sum_{y_j} p(y_j \mid r)\sum_{d_j} p(d_j \mid y_j)p(x_j \mid d_j)\Big)$$

Rather than marginalizing over all choices of $d$, we restrict each sum to a single term. For $d_i$, we choose the most likely detector output $d_i = d_i^*$. For $d_j$, we choose the corresponding $y_j$ in the outer sum. Together, these simplifications reduce the total number of unnecessary LM queries about unlikely object confusions, and give a lower bound:

$$\geq p(d_i = d_i^* \mid y_i)p(x \mid d_i = d_i^*)\sum_r p(r)(p(y_i \mid r)\Big(\prod_j \sum_{y_j} p(y_j \mid r)p(d_j = y_j \mid y_j)p(x_j \mid y_j)\Big)$$

Applying Bayes' rule locally:

$$= \frac{p(y_i \mid d_i = d_i^*)p(d_i = d_i^*)}{p(y_i)}\frac{p(d_i = d_i^* \mid x_i)p(x_i)}{p(d_i = d_i^*)}$$
$$\sum_r p(r)p(y_i \mid r)\Big(\prod_i \sum_{y_j}\frac{p(r \mid y_j)p(y_j)}{p(r)}p(d_j = y_j \mid y_j)\frac{p(d_j = y_j \mid x_j)p(x_j)}{p(d_j = y_j)}\Big)$$

Finally, we make two modeling assumptions. First, we assume that of the form $p(y)$ and $p(d)$—the marginal distributions of true and noisy object labels—are uniform. This allows us to use LMs as a source of information about object co-occurrence probabilities without relying on their assumptions about base class frequency. Second, for non-target detections $x_j$, we assume the probability that noisy labels match the true labels is constant over object categories. Then, dropping constant terms gives:

$$\propto p(y_i \mid d_i = d_i^*)p(d_i = d_i^* \mid x_i)\sum_r p(r)p(y_i \mid r)\Big(\prod_j \sum_{y_j}\frac{p(r \mid y_j)}{p(r)}p(d_j = y_j \mid x_j)\Big)$$

## A.2 Socratic Model Baseline

We attempt to make the Socratic models inference procedure for this task as analogous to the LaMPP approach as possible: the LM must account for both room-object co-occurrence likelihoods and object-object resemblance likelihoods when predicting true labels. However, here, the LM must

| Base Model | mIoU | ΔIoU by Object Category |
|---|---|---|
| RedNet (ID) | 47.8 | |
| + LAMPP | 48.3 | |
| + SM | 37.5 | |
| RedNet (OOD) | 33.8 | |
| + LAMPP | 34.0 | |
| CLIPSeg (ZS) | 27.7 | |
| + LAMPP | 28.2 | |

Table 4: Image Segmentation results. We report mIoU, or intersection-over-union averaged over each object category. We also report the improvements to IoU for each object category when LAMPP or Socratic Models is applied to RedNet trained on in-domain or out-of-domain data, and CLIPSeg applied zero shot.

implicitly incorporate these likelihoods into its text-scoring, rather than integrating them into a structured probabilistic framework.

Specifically, the Socratic model baseline is given model predictions $\widehat{d}_i$ for *each segment* $x_i$ and re-labels each segment by querying GPT-3 with:

> *You can see:* $[\widehat{\underline{d}}]$
>
> *You are in the* $[r]$
> *The thing that looks like* $[\widehat{d}_i]$ *is actually* $[y_i]$ .

The LM is given the non-highlighted portions and asked to generate the portions highlighted in yellow. $\widehat{\underline{d}}$ is the set of all unique objects detected by the base model, written out as a comma-separated list. $r$ is a room type generated by the LM based on these objects (inferred by normalizing over possible room types), and $y_i$ is the actual identity of the object corresponding to this segment. We replace all pixels formerly predicted as $\widehat{d}_i$ with $y_i$.

## A.3 FULL RESULTS

See Table 4 for the full results over object categories. In the ID case, note that shower curtain improved dramatically with the incorporation of LAMPP. This is because the base model almost always classifies shower curtains as window curtains, while explicitly incorporating room priors from language models fixes this major class of error. Moreover, in the OOD setting, the base image model sees far fewer examples of nightstands and consequently never predicts nightstands on the test data. (Nightstands are frequently predicted to be tables and cabinets instead). This is likewise rectified with LAMPP: background knowledge from language reduces model sensitivity to a systematic bias in dataset construction

The goal objects that improved the most with the incorporation of LAMPP (shower curtain, nightstand in the OOD case) are all ones that strongly co-occur with specific rooms, indicating the usefulness of incorporating room-object co-occurrence priors.

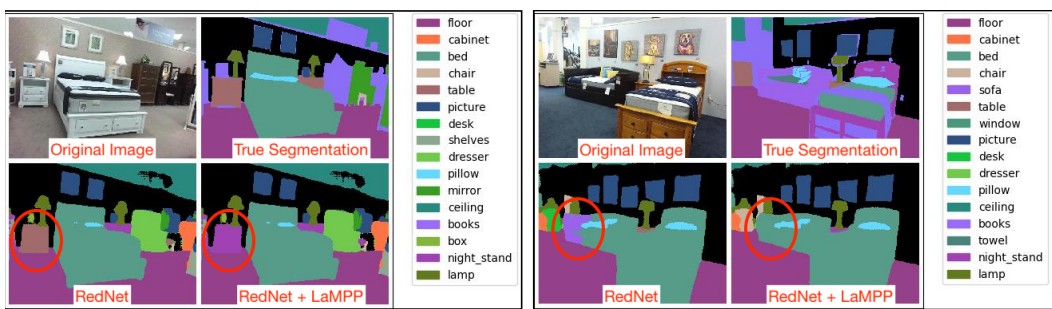

(a) Table → Nightstand                    (b) Sofa → Bed

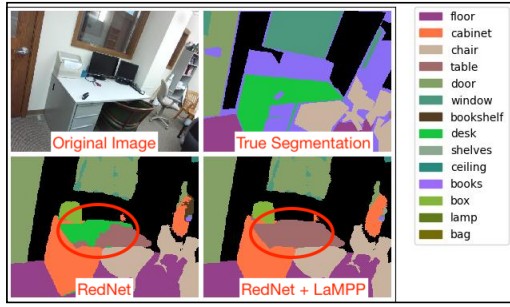

(c) Desk → Table

Figure 5: Image Segmentation Domain: Examples of errors from LAMPP.

## A.4 ERROR ANALYSIS

We manually inspect 50 random outputs from LAMPP on the ID setting and identify broadly the following categories of errors, which are illustrated in Fig. 5.

1. Mislabelling: The ground truth labels are sometimes incorrect. For example, in Fig. 5(a), the nightstand is mislabelled as a "table". LAMPP is then wrongly penalize when it changes the table to a nightstand.

2. Missegmentation: Object boundaries may be incorrectly identified, as in Fig. 5(b) and Fig. 5(c). This can confuse LAMPP as it may be unlikely for two visually-similar items to co-occur in the same room (e.g. beds and sofas), and LAMPP may try to "paint over" the mis-segmented chunk with the wrong label.

3. Example inconsistent with LM priors: For a small number of examples, the LM prior may be incorrect. These examples can be thought of as being sampled from the tail of the distribution. For example, in Fig. 5(b), the LM believes that it is unlikely for beds and sofas to co-occur in the same room. However, this particular example is of a furniture shop, meaning that it is actually the case that the bed and sofa are next to each other.

4. Insufficient information to LM: Because we may only interact with LMs through text, the LM loses out on certain visual signal that may help disambiguate between two semantically-similar object labels. For example, in Fig. 5(c), LAMPP incorrectly converts the desk into a table, likely because the LM does not have sufficient information to conclude that the image is of an office setting (desks are more often used to in the context of work). Note that "computer" and "printer" weren't available labels for the segmentation model, but may have be useful information for the LM.

# B    LaMPP FOR NAVIGATION

## B.1    SOCRATIC MODEL BASELINE

As in the image segmentation case, we have a Socratic model baseline. LM priors are integrated into exploration through directly querying the LM with

> *The house has:* [$r$].
> *You want to find a* [$g$]. *First, go to each*  [$r_0$] . *If not found, go to each*  [$r_1$] . *If not found, go to each* $\cdots$

whereby $r$ is a list of all room types in the environment, for example, *3 bathrooms, 1 living room, 1 bedroom*. The LM returns the best room type $r_0$ to navigate to in order to find $g$. The agent visits all $r_0$ in order of proximity. If the object is not found, the LM is queried for the next best room type to visit, etc., until the object is found or we run out of rooms in the environment.

## B.2    METHODS

Below we provide intuition for why our policy outlined in Section 4.1 follows from the generative story of the success score $y_i$ depicted in Fig. 3. We assume objects are equally likely to be distributed across all rooms.

At a high level, an agent greedily optimizing for success should move to a room $r$ that maximizes $p(y \mid x, r, g)$ which, following Fig. 3 is equivalent to maximizing:

$$p(y \mid g, r)p(x \mid y) \propto p(y \mid g, r)p(y \mid x)p(x)/p(y) \propto p(y \mid g, r)p(y \mid x) \tag{6}$$

The policy that maximizes Eq. (6) iterates between *navigation* and *selection* actions:

1. *Navigation*: We navigate to the unobserved room with the highest $p(y \mid r)$.[6] This follows directly from Eq. (6) and the assumption of uniform initial object locations.

2. *Selection*: Once we have some observations $x$ of the room, we branch off into two cases:

   (a) *Deciding to continue:* We do not see $g$ (low $p(y \mid x)$). If we take $n$ (= 25) steps in the room and do not see any instance of $g$, we become relatively confident that the room does not contain $g$ at all, so $p(y \mid x) \to 0$ for that room. (Consequently, the objective $p(y \mid x)p(y \mid r)$ also $\to 0$ for this room.) We eliminate this room from the list of unobserved rooms and return to step 1.

   (b) *Deciding to stop:* We see $g$ (high $p(y \mid x)$), either on the way to the room or within the room. We navigate to exact location that maximizes $p(y \mid x)$ and decide to stop based on $p(y \mid x)p(r \mid y)$.

## B.3    FULL RESULTS

See Table 5 for the full set of results over goal-object categories. Note that the goal objects that improved the most with the incorporation of LaMPP (TV monitor, sofa, toilet) are all ones that strongly co-occur with specific rooms, indicating the usefulness of incorporating room-object co-occurrence priors.

## B.4    UNIFORM PRIORS MODEL VS. ORIGINAL STUBBORN AGENT MODEL

Recall that we modified the original STUBBORN agent in the navigation task to utilize the high-level map, by uniformly sampling a random (unvisited) room to visit. More specifically, the uniform priors model utilizes effectively the same high-level policy as LaMPP-based agent, but replaces LM priors over over object-room co-occurrences with uniform priors:

$$p(y = \text{True} \mid r, g) = \frac{1}{\text{\# room types in environment}}. \tag{7}$$

---

[6]For rooms with the same $p(y \mid r)$, or if there are multiple rooms of the same type, we navigate in order of proximity to the agent's current location.

| Base Model | SR | | SPL | | $\Delta$SR by Goal Object Category |
|---|---|---|---|---|---|
| | Class | Freq. | Class | Freq. | |
| Orig STUBBORN | 52.7 | 53.8 | 21.2 | 22.8 | |
| Unif Prior (ZS) | 52.1 | 51.7 | 20.2 | 21.5 | |
| + LaMPP | 66.5 | 65.9 | 35.4 | 36.0 | |
| + SM | 61.2 | 65.3 | 31.3 | 35.1 | |

Table 5: Navigation results, showing *class-averaged* success rate (SR; averaged over goal objects) and *frequency-averaged* success rate (averaged over episodes). We also report SPL, Success weighted by (normalized inverse) Path Length. Finally, we report the improvements to success rate over each object category when LaMPP vs. Socratic Models is applied.

| Model | Class-Avg. SR | Freq.-Avg. SR |
|---|---|---|
| LaMPP | 66.5 | 65.9 |
| $-p(y \mid r)$ during selection | 58.8 | 64.9 |
| Socratic model | 61.2 | 65.3 |

Table 6: Navigation results verification ablations. We ablate the LM uncertainties over $p(y \mid r)$ when computing the selection action, making LaMPP functionally similar to a Socratic model baseline. We find that having these uncertainties are crucial; without them, LaMPP actually *underperforms* the Socratic model baseline.

Note in the zero-shot case we have no additional information about $p(y \mid r, g)$, so we must assume it is uniform.

We report average success rates over object categories of the original STUBBORN agent vs. a uniform priors agent vs. an LaMPP-based agent vs. a Socratic-model-based agent in Table 5. We find that LaMPP is able to outperform both the original STUBBORN agent and the uniform priors agent.

### B.5 ADDITIONAL ANALYSIS

Why does LaMPP outperform Socratic modeling? In the SM approach, high-level decisions from the LM and low-level decisions from observation models are usually considered separately and delegated to different phases (it is hard to combine these information sources in string-space): in our implementation, the SM baseline uses the top-down LM for *navigation*, and the bottom-up observation model for *selection*.

Because the policy dictated by the LaMPP probabilistic model also ignores bottom-up observation probabilities until the goal object is observed, the *navigation* step of both approaches is functionally equivalent. However, for the *selection* step, we find that combining bottom-up and top-down uncertainties is crucial (recall that LaMPP thresholds $p(y \mid x, r, g)$ at selection steps, which decomposes to $p(y \mid x)p(y \mid r, g)$).

To further understand and how using LM probabilities contributes at this phase, we run a version of LaMPP where we simply change the decision rule at the selection action to $p(y \mid x)$. Results are reported in Table 6. Note that we actually *underperform* the SM baseline when we take away top-down uncertainties $p(y \mid r, g)$ — once again highlighting the importance of combining both sources of uncertainty.

## C LaMPP FOR VIDEO-ACTION SEGMENTATION

| Base Model | Step Recall | | Δ Step Recall by Task |
|---|---|---|---|
| | Class | Freq. | |
| HSMM (ZS) | 44.4 | 46.0 | |
| + LaMPP | 45.7 | 47.9 | |
| HSMM (OOD) | 37.6 | 40.9 | |
| + LaMPP | 38.1 | 41.2 | |

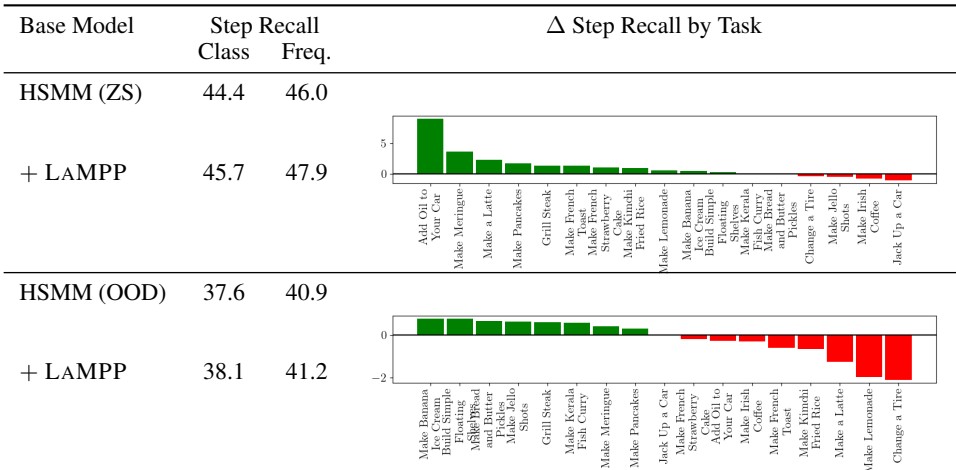

Table 7: Action recognition and segmentation results, showing *class-averaged* step recall (averaged over tasks) and *frequency-averaged* step recall (averaged over videos). We also report the improvements to step recall over each object category when LaMPP is applied.

## C.1 SELECTING THE LM PROMPT

We performed significant prompt engineering in this setting (on the training set), and found that with all of the prompts we tried, GPT-3 was biased towards outputting actions in the order that they were presented to the model. The following are some examples of other prompts we tried:

> *Your task is to* [$t$]. *Your actions are: {*[$Y$]*}*
> *The step after* [$y$] *is* [$y'$]: [*plausible / implausible*]

> *Your task is to* [$t$]. *Your actions are: {*[$Y$]*}*
> *The step after* [$y$] *is* [$y'$]

We also tried having the LM produce a global ordering and setting probability from an action later in the sequence to earlier in the sequence to 0:

> *Your task is to* [$t$]. *Your set of actions is: {*[$Y$]*}*
> *The correct ordering is:*

The prompt in 5.1 was slightly better at combating this effect, though still highly imperfect.

## C.2 FULL RESULTS

See Table 7 for the full set of results over actions.

## C.3 ERROR ANALYSIS

In this domain, GPT-3 often produced priors that were unaligned with the true distribution, leading to errors in downstream prediction with LaMPP. As noted in C.1, we found that GPT-3 has a proclivity to output actions in the order that they were presented in the prompt. To further probe errors in GPT-3's priors, we plot GPT-3's distribution over actions for tasks against the ground-truth distribution over actions in the real training and validation datasets. We found that, when these priors were uncorrelated with real priors from the dataset, there tended to fall into the following categories:

1. Bias towards ordering in prompt: see discussion in Appendix C.1.
2. Task ambiguity: The LM and ground-truth demonstrations may prefer slightly different ways of accomplishing the same task. For example, in the "Grill steak" task (Fig. 6(a,b)),

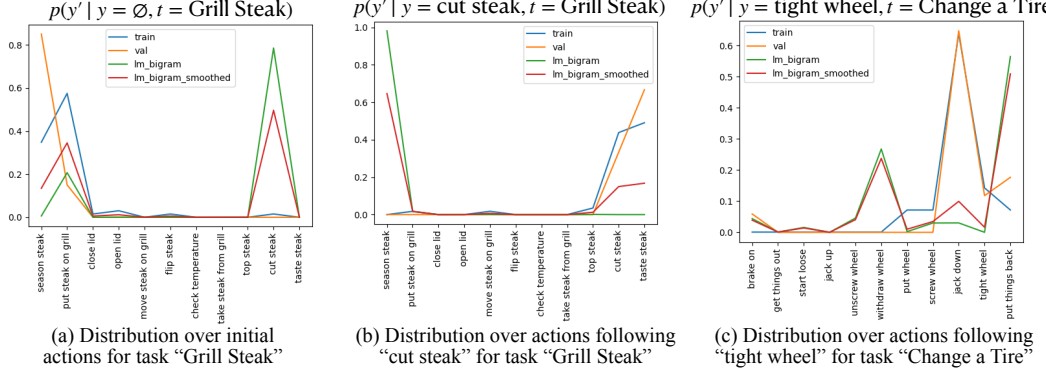

(a) Distribution over initial actions for task "Grill Steak"

(b) Distribution over actions following "cut steak" for task "Grill Steak"

(c) Distribution over actions following "tight wheel" for task "Change a Tire"

Figure 6: LM vs. ground-truth distributions over action sequences for tasks. GPT-3 priors are plotted in blue, while training-set distributions and validation-set distributions are plotted in blue and orange respectively. The red line represents a "smoothed" LM prior which combines GPT-3 priors with learned priors.

GPT-3 prefers to cut the steak *before* cooking (green line in Fig. 6(a) shows that the LM's highest-probability initial action is "cut steak"), while the ground-truth distribution prefers cutting the steak *after* cooking (blue & orange lines in Fig. 6(b) shows that the datasets' highest-probability action after "cut steak" is "taste steak").

3. Action ambiguity: Note that, in this dataset, actions are typically named with a very short description, which may sometimes cause ambiguity. For example, in the "Change a tire" task Fig. 6(c), it is unclear whether actions involving the wheel is referring to the old wheel or the new wheel, and the action "start loose" is unclear.

4. Structural limitations of HSMM: Because we are modelling a sequence of scenes from videos with an HSMM, the action segmentations may be imperfect – for example, in Fig. 6(b), the HSMM learns that it is highly likely for the action after "cut steak" to also be "cut steak" from the training datasets (with probability $\sim 0.5$), perhaps because the cutting scene hasn't ended. The LM is unable to these transitions.

5. Wrong LM Priors: GPT-3 sometimes simply has incorrect priors about the action orderings for tasks, for example putting a fairly high probability on "widthdraw wheel" after "tight wheel" for task "Change a tire" in Fig. 6(c).

# D TRAINING DETAILS

## D.1 HYPERPARAMETERS

Since our method builds upon each base model, we used standard hyperparameters to train and run inference for each base model. For the components of LAMPP that we build on top of the base models, there are usually only a few hyperparameters to tune, which we tune based on performance on a held-out development set.

In image segmentation, we tune is a weighting term $\lambda$ at the final decision layer that trades off the weight on the LM prior and the observation posterior.

$$\log p(y \mid x) = \lambda \log p(y) + (1 - \lambda) \log p(x \mid y)$$

We find he optimal $\lambda$ to be $0.9$. In the navigation case, we tune a threshold $\tau$ that determines when the agent stops ($p(y \mid x, r, g) > \tau$) and find the optimal $\tau$ is $0.2$ For video-action segmentation, we have a hyperparameter $\alpha$ controlling the weight on the prior (see Eq. (4)), which we set to $10$.

# E COMPUTATIONAL DETAILS

We run all experiments on a single 32GB NVIDIA V100 GPU.

For image segmentation experiments, we used either: 1. a pretrained RedNet checkpoint, 2. a RedNet checkpoint trained for 1000 epochs on OOD data, 3. a pretrained zero-shot CLIPSeg model. 1 and 3 required no training, 2 required training for $\sim$ 3-4 days on the GPU. Inference for a single experiment takes several hours each on a single GPU.

For navigation experiments, we used the pre-trained STUBBORN model. A full inference run on our GPU takes about a full day to complete. (However, multiple inference experiments could be run in parallel using the same GPU – the bottleneck is simulating every step of every trajectory, rather than GPU inference.)

For video-action segmentation experiments, we use the HSMM model trained by Fried et al. (2020). On a single GPU, the model typically takes only a few minutes to fully train and a few minutes for inference. Several inference experiments can be run in parallel on the same GPU.

Finally, to derive LM priors, we query the OpenAI API. Using our method uses a budget of $\sim$\$12 (in totality, over *all experiments in all domains*), as it queries a fixed number queries ahead of time ($\sim$4k queries), which get reused over the entire test set. Meanwhile, running full SM experiments requires on the order of several hundred dollars in totality – frequently requiring multiple queries *per test sample*.

## F    LIMITATIONS

While flexible, LAMPP is not applicable to every machine learning task, and specifically requires (1) a formalization of the task as a probabilistic graphical model, in which (2) values of some variables (e.g. labels) can be represented as natural language strings, and (3) probabilistic relations between these variables are described (with some degree of precision) in large-scale text corpora. As shown above, many success cases involve common-sense knowledge about human environments or human task structures; more specialized tasks (e.g. fine-grained image classification or medical diagnosis) may not enjoy the same benefits.

