# OpenReview forum: "LaMPP: Language Models as Probabilistic Priors for Perception and Action"
_ICLR.cc/2024/Conference — Submitted to ICLR 2024_

### Official Review · Reviewer_xpjw · 2023-10-25

**Soundness:** 2 fair
**Presentation:** 3 good
**Contribution:** 2 fair
**Rating:** 6
**Confidence:** 3

**Summary:**

The authors present a method for incorporating LLM likelihoods into tasks such as object segmentation, navigation, and action recognition. For each task, a task-specific base model is combined with a pre-trained LLM and each model estimates the likelihood of an event given some context, e.g. the likelihood of a bed in a bedroom and then likelihood that the given pixels are a bed.The LLM is used to refine the base model's estimated likelihood for an event and in doing so incorporates semantic background knowledge not necessarily present in the task-specific dataset into the final decision. The approach is primarily compared to the Socratic Method for prompting LLMs and the task-specific models run in the absence of the LLM. Three tasks are assessed and for each task the models are assessed on OOD, in-distribution, and zero-shot settings. Across tasks and settings, the proposed approach LaMPP out performs other methods.

**Strengths:**

- The paper is well written and easy to follow.
- The authors introduce an interesting way to incorporate background knowledge from LLMs into graphical models to estimate the likelihood of events.
- The authors evaluate on a tasks with different degrees of temporal reasoning required.

**Weaknesses:**

- The authors primarily assess the model on tasks where the LLM is expected to perform well. For example, the task-specific training dataset does not contain all possible combinations of objects and the LLM is able to overcome this because of the large amount of data it was trained on. The authors do not assess performance on cases where the LLM may perform poorly on the downstream task. For example, sofas and sinks being in the same room or complex spatial reasoning tasks. The ways in which the LLM could harm performance are not explored and assessed. This is touched on at the very end of the conclusion, but should be addressed and measured more centrally.

**Questions:**

- What is mean by "easy-to-predict object labels"? What makes them easy? What about hard object labels?
- For the LM Queries in the semantic segmentation task, where does "r" come from?
- How well is the model able to account for likelihoods of different numbers of objects? For example, a bedroom is likely to have a bedside table, but not likely to have 12 of them.
- For 3.2 Experiments, you mention that "a RedNet checkpoint" is used. How was this checkpoint selected?
- What are the differences in compute time between LaMPP, SM, and the base models?

---

> ### Author Response · Authors · 2023-11-16
>
> ## Weaknesses
> *W1: The authors primarily assess the model on tasks where the LLM is expected to perform well. For example, the task-specific training dataset does not contain all possible combinations of objects and the LLM is able to overcome this because of the large amount of data it was trained on. The authors do not assess performance on cases where the LLM may perform poorly on the downstream task. For example, sofas and sinks being in the same room or complex spatial reasoning tasks. The ways in which the LLM could harm performance are not explored and assessed. This is touched on at the very end of the conclusion, but should be addressed and measured more centrally.*
>
> Our paper rests on the key insight that LLMs, being pre-trained on large amounts of data, contain generic priors that are relatively well-aligned with many real tasks. Indeed, empirically, we found that LMs generally contained priors that were useful for real tasks (note we did not specifically design these tasks in a way to be conducive for LLMs – rather these were taken from real datasets), and that these priors offer complementary benefits to what domain-specific vision/robotics/etc. models have learned through supervised training.
>
> Furthermore, we also anticipate that as LLMs improve, they will have even better priors covering a broader range of tasks. This has the potential to include priors for some complex spatial reasoning tasks – indeed we see large improvements on these sorts of tasks moving from GPT3 to GPT4.
>
> Finally, the reviewer makes a good suggestion that we should provide examples of when LLM priors fail. The following is an example where the LLM has an incorrect prediction.
>
> > Domain: Image Semantic Segmentation
> >
> > LM prior: predicts that the object next to the bed with a lamp on it is likely to be a **nightstand**
> >
> > Real label: the object is actually a **dresser**
>
> We will include more qualitative examples of failure cases in the appendix of future versions of the paper.
>
>
>
>
> ## Questions
> *Q1: What is mean by "easy-to-predict object labels"? What makes them easy? What about hard object labels?*
>
> By “easy-to-predict object labels”, we mean the ones that the domain-specific base model already achieves high accuracy on, e.g. object classes that are frequent in the training set or can be recognized unambiguously without scene-level context..
>
> *Q2: For the LM Queries in the semantic segmentation task, where does "r" come from?*
>
> Rooms $r$ are latent / unobserved (note that they are unshaded in the figure). This means that to derive the probability of a pixel label, we perform inference and marginalize over all possible rooms.
>
> *Q3: How well is the model able to account for likelihoods of different numbers of objects? For example, a bedroom is likely to have a bedside table, but not likely to have 12 of them.*
>
> We didn’t find this necessary to model in our experiments, but could be straightforwardly accommodated with a slightly different graphical model.
>
> *Q4: For 3.2 Experiments, you mention that "a RedNet checkpoint" is used. How was this checkpoint selected?*
>
> For ID generalization, we used a pre-trained checkpoint released by https://github.com/JindongJiang/RedNet . For OOD generalization, we fixed the number of training epochs to 1000 and selected the last RedNet checkpoint after training finished, without further hyper-parameter tuning. These details can be found in Appendix E.
>
> *Q5: What are the differences in compute time between LaMPP, SM, and the base models?*
>
>
> Good question! LaMPP has a fixed overhead cost for querying the GPT3 API, but otherwise runs the exact same inference procedure as the base model, adding a relatively inexpensive probabilistic inference step on top. SM adds the most computation time, as it has to query GPT3 (potentially multiple times) for inference on every example. We time inference over a subsample of 100 examples in image semantic segmentation and report results below. We will also insert these results into future versions of the paper:
>
> 1. Base Model: 1.21 s/it
> 2. SM: 40.7 s/it
> 3. LaMPP: 1.83 s/it

---

> ### Author Response · Authors · 2023-11-20
>
> Dear Reviewer xpjw,
>
> Thank you for your detailed review and feedback. As the discussion period is coming to a close, please let us know if our response has adequately addressed your concerns, or if you have any remaining questions and concerns. If not, we would appreciate if you could raise your score. Thank you for your hard work!

---

> > ### Comment · Reviewer_xpjw · 2023-11-21
> > **Thank you to author response**
> >
> > Thank you for your responses.
> >
> > If the paper is accepted, it would be beneficial to include a discussion of current limitations where the LLM’s priors are subpar or harmful to help push and advance next steps in research.

---

> > > ### Author Response · Authors · 2023-11-22
> > > **Please check revision**
> > >
> > > Dear reviewer xpjw,
> > >
> > > Thank you for your feedback! Please note that our original submission already had a brief limitations section in Appendix F. However, we have fleshed out the discussion of limitations by adding a concrete error analysis in Appendix A.4 (on image segmentation) and Appendix C.3 (on video-action segmentation), the two domains where we saw comparatively less improvement. Please check the revised PDF. We have also copied the text from those sections below:
> > >
> > > **Appendix A.4**
> > > > We manually inspect 50 random outputs from LAMPP on the ID setting and identify broadly the
> > > following categories of errors, which are illustrated in Fig. 5.
> > > > 1. Mislabelling: The ground truth labels are sometimes incorrect. For example, in Fig. 5(a),
> > > the nightstand is mislabelled as a “table”. LAMPP is then wrongly penalize when it changes
> > > the table to a nightstand.
> > > > 2. Missegmentation: Object boundaries may be incorrectly identified, as in Fig. 5(b) and
> > > Fig. 5(c). This can confuse LAMPP as it may be unlikely for two visually-similar items to
> > > co-occur in the same room (e.g. beds and sofas), and LAMPP may try to “paint over” the
> > > mis-segmented chunk with the wrong label.
> > > > 3. Example inconsistent with LM priors: For a small number of examples, the LM prior may
> > > be incorrect. These examples can be thought of as being sampled from the tail of the
> > > distribution. For example, in Fig. 5(b), the LM believes that it is unlikely for beds and
> > > sofas to co-occur in the same room. However, this particular example is of a furniture shop,
> > > meaning that it is actually the case that the bed and sofa are next to each other.
> > > > 4. Insufficient information to LM: Because we may only interact with LMs through text, the
> > > LM loses out on certain visual signal that may help disambiguate between two semantically-
> > > similar object labels. For example, in Fig. 5(c), LAMPP incorrectly converts the desk into
> > > a table, likely because the LM does not have sufficient information to conclude that the
> > > image is of an office setting (desks are more often used to in the context of work). Note that
> > > “computer” and “printer” weren’t available labels for the segmentation model, but may have
> > > be useful information for the LM.
> > >
> > > **Appendix C.3**
> > > > In this domain, GPT-3 often produced priors that were unaligned with the true distribution, leading
> > > to errors in downstream prediction with LAMPP. As noted in C.1, we found that GPT-3 has a
> > > proclivity to output actions in the order that they were presented in the prompt. To further probe
> > > errors in GPT-3’s priors, we plot GPT-3’s distribution over actions for tasks against the ground-truth
> > > distribution over actions in the real training and validation datasets. We found that, when these priors
> > > were uncorrelated with real priors from the dataset, there tended to fall into the following categories:
> > > > 1. Bias towards ordering in prompt: see discussion in Appendix C.1.
> > > > 2. Task ambiguity: The LM and ground-truth demonstrations may prefer slightly different ways of accomplishing the same task. For example, in the “Grill steak” task (Fig. 6(a,b)), GPT-3 prefers to cut the steak before cooking (green line in Fig. 6(a) shows that the LM’s highest-probability initial action is “cut steak”), while the ground-truth distribution prefers cutting the steak after cooking (blue & orange lines in Fig. 6(b) shows that the datasets’ highest-probability action after “cut steak” is “taste steak”).
> > > > 3. Action ambiguity: Note that, in this dataset, actions are typically named with a very short description, which may sometimes cause ambiguity. For example, in the “Change a tire” task Fig. 6(c), it is unclear whether actions involving the wheel is referring to the old wheel or the new wheel, and the action “start loose” is unclear.
> > > > 4. Structural limitations of HSMM: Because we are modelling a sequence of scenes from videos
> > > with an HSMM, the action segmentations may be imperfect – for example, in Fig. 6(b), the
> > > HSMM learns that it is highly likely for the action after “cut steak” to also be “cut steak”
> > > from the training datasets (with probability ∼ 0.5), perhaps because the cutting scene hasn’t
> > > ended. The LM is unable to these transitions.
> > > > 5. Wrong LM Priors: GPT-3 sometimes simply has incorrect priors about the action orderings
> > > for tasks, for example putting a fairly high probability on “widthdraw wheel” after “tight
> > > wheel” for task “Change a tire” in Fig. 6(c)
> > >
> > > We understand that this is very short notice, but we would appreciate if you could take a look and let us know if you have remaining questions. If this has satisfactorily addressed your concerns, we would also appreciate if you could raise your score. Thank you!

---

### Official Review · Reviewer_WR1X · 2023-10-30

**Soundness:** 3 good
**Presentation:** 3 good
**Contribution:** 3 good
**Rating:** 6
**Confidence:** 3

**Summary:**

The paper presents the use of LM (language model) as a source of prior distributions over labels, decisions or model parameters. The approach is empirically demonstrated through 3 different domains -- semantic image segmentation using SUN RGB-D dataset, indoor object navigation using Habitat challenge, and action recognition on Cross-Task dataset.

The three case studies cover diverse objectives and show consistent improvement on in-distribution, out-of-distribution and zero shot generalization. Further, the studies also show that the proposed approach is more cost effective than the existing approaches such as Socratic Modeling.

**Strengths:**

-- The empirical studies show a delta between the proposed and existing approaches for the in-distribution results for image semantic segmentation.

-- The three domains chosen, while having some similarities in terms of input modalities, is sufficiently diverse to demonstrate the generalization of the approach.

**Weaknesses:**

-- The empirical results for zero-shot and OOD show very minor improvements relative to baseline. For image semantic segmentation, the results in A.3 show 0.2 and 0.5 mIOU improvement for OOD and ZS resp. Similarly, the improvements for ZS and OOD for video segmentation are not very significant.

-- The general approach seems to require a significant amount of domain specific information which might work for smaller / restricted domains but will face issues when generalizing to broader / open domains.

**Questions:**

-- Can this approach be shown to work on a large dataset (e.g., 100k+ labels)?

---

> ### Author Response · Authors · 2023-11-16
>
> Thank you for your feedback! Below we address the weaknesses and questions.
>
> ## Weaknesses
> ### W1: The empirical results for zero-shot and OOD show very minor improvements relative to baseline. For image semantic segmentation, the results in A.3 show 0.2 and 0.5 mIOU improvement for OOD and ZS resp. Similarly, the improvements for ZS and OOD for video segmentation are not very significant.
>
> We agree that on average the improvements are small in the image segmentation and video segmentation domains, but this paper focuses on improvements in the long tail of the distribution. The improvements are quite significant for specific categories in all 3 image segmentation cases (+18.9 in ID, +8.92 in OOD, +13.3 ZS), without much loss to the other categories (no more than -2.16 in ID, -2.5 in OOD, -5.0 ZS). The same can be said of the zero-shot video segmentation case, where “add oil to car” improves by +8.96 points, while the category that drops the most (“jack up a car”) drops by -1.03 points. Note also that almost all categories improve in the navigation domain.
>
> ### W2: The general approach seems to require a significant amount of domain specific information which might work for smaller / restricted domains but will face issues when generalizing to broader / open domains.
>
> LaMPP generally requires: 1. designing a graphical model for a task, and 2. designing prompts for extracting priors for LMs to insert into the graphical model.
>
> 1. We offer a guideline for designing the graphical model in section 2: you can either decompose the prediction task into $p(y)p(x|y)$, using the LM to put a prior over labels $p(y)$, or you can decompose learning into $p(theta)p(y|x, theta)$, and use the LM to put a prior over parameters $p(theta)$. While there may be some customization necessary to work out the exact form of each of these probabilities for each task, the general structure of graphical models follows from these above design decisions (figures 2–4).
> 2. Defining prompts (prompt engineering) is a typical requirement for inducing LMs to perform various tasks these days. Thus, at this step our method requires no more domain-specific engineering than what is typically required.
>
> Note that alternative methods for incorporating LM priors into domain-specific grounded models all require some degree of domain-specific engineering. For example, Socratic models / model chaining / tool usage require that the prompts and associated model chains are designed on a per-task basis. Nonetheless, these paradigms have proven useful and transferable. In fact, LaMPP offers a further step in the direction of standardization and formalization by offering a unifying view of LMs as priors. We will try to include more of this “philosophical” motivation in future versions of the paper.

---

> ### Author Response · Authors · 2023-11-20
>
> Dear Reviewer WR1X,
>
> Thank you for your detailed review and feedback. As the discussion period is coming to a close, please let us know if our response has adequately addressed your concerns, or if you have any remaining questions and concerns. We appreciate your hard work!

---

### Official Review · Reviewer_C44C · 2023-11-01

**Soundness:** 3 good
**Presentation:** 3 good
**Contribution:** 2 fair
**Rating:** 5
**Confidence:** 4

**Summary:**

I have reviewed a previous version of this manuscript.

The manuscript formulates labeling and decision-making as inference in probabilistic graphical models, where language models can act as the probabilistic prior distribution over labels, decisions, and parameters. The manuscript takes a case study approach, to consider different task settings.

**Strengths:**

The manuscript is well-written.

The problem formulation is interesting.

**Weaknesses:**

Figure 1 is missing illustration for the “robot navigation” task, probably due to attempts to keep it in a floating single column format. Consider redrawing the figure.

Section {3,4,5} (prompting): Provide discussion of how the fixed text prompt templates were chosen, particularly for the action recognition case study. Provide some examples of other templates that were tried, even if they did not prove successful.

Section 5.2: The lack of discussion surrounding evaluation on ID settings (still) stands out; it would be a worthwhile a comparison of task/domain difficulty. Provide discussion in the main content.

Section 5.3: Several other LLMs and VLMs have been available for quite some time now. Missing comparisons with other foundation models.

Section 6: The related works section is sparse and uninformative. The manuscript forgot to highlight *both* the strengths and weaknesses of the relevant related work and describe the ways in which the proposed method improves on (or avoids) those same limitations. This should serve as a basis for the experimental comparisons and should follow from the stated claims of the paper.

**Questions:**

N/A – see above

---

> ### Author Response · Authors · 2023-11-16
>
> Thank you for your feedback! Below we address the weaknesses. We will note that W1, W2, W3, and W5 all suggest adding additional content to our paper, which is currently at the page limit. We will try our best to compress our other content to accommodate any necessary changes, but please also let us know if there is anything you believe can be deleted!
> ## Weaknesses
> ### W1: Figure 1 is missing illustration for the “robot navigation” task, probably due to attempts to keep it in a floating single column format. Consider redrawing the figure.
>
> Thanks for the feedback, we will redraw the figure to include the robot navigation task.
>
> ### W2: Section {3,4,5} (prompting): Provide discussion of how the fixed text prompt templates were chosen, particularly for the action recognition case study. Provide some examples of other templates that were tried, even if they did not prove successful.
>
> Thanks for the feedback, we will include this discussion in future versions of the paper. For the image semantic segmentation and navigation tasks, there was minimal prompt engineering required – the LM outputted well-calibrated priors, extractable with the simplest prompts.
> However, for video-action recognition we had to do a substantial amount of prompt engineering. For example, some other prompts we tried that included:
>
> > **Prompt:** Your task is to *[t]*. Your actions are: {*[Y]*}
> >
> > The step after *[y]* is *[y’]*:
> >
> > **LM:** [plausible / implausible]
>
> > **Prompt:** Your task is to *[t]*. Your actions are: {*[Y]*}
> >
> > The step after *[y]* is
> >
> > **LM:** [y’]
>
> We also tried having the LM produce a global ordering and setting probability from an action later in the sequence to earlier in the sequence to 0:
>
> > **Prompt:** Your task is to *[t]*. Your set of actions is: {*[Y]*}
> >
> > The correct ordering is:
>
>
> ### W3: Section 5.2: The lack of discussion surrounding evaluation on ID settings (still) stands out; it would be a worthwhile a comparison of task/domain difficulty. Provide discussion in the main content.
>
> Good catch– we did not do in-domain evaluation for the video-action task as it did not improve performance. We will add a small discussion about this in the main text of future versions. We believe this is because the base model is simple enough that simply counting transitions and observation co-occurrence is sufficient for estimating parameters, and the action space is small enough that the data is sufficient to comprehensively cover actions and transitions for each task. Thus, the fully-supervised model is as good as it will get.
>
> ### W4: Section 5.3: Several other LLMs and VLMs have been available for quite some time now. Missing comparisons with other foundation models.
>
> We have comparisons to three different vision models across three different tasks, including comparison to both CLIPSeg and RedNet in segmentation tasks. If you have a specific comparison you would like to see, please let us know. Furthermore, our method is complementary to the specific foundation model used for a task, and can operate on top of any base foundation model. While VLMs may be starting to find some success with simpler tasks like image segmentation, we anticipate that it will be a while before multimodal models can accommodate actions and videos.
>
> ### W5: Section 6: The related works section is sparse and uninformative. The manuscript forgot to highlight both the strengths and weaknesses of the relevant related work and describe the ways in which the proposed method improves on (or avoids) those same limitations. This should serve as a basis for the experimental comparisons and should follow from the stated claims of the paper.
>
> Thanks for the feedback. Again, we believe we have highlighted strengths and weaknesses of the most pertinent related work: Socratic models (L380-384) and LMPriors (L385-387). Due to space limitations, we were unable to fit a more comprehensive related works section, but we will expand it in future versions.

---

> ### Author Response · Authors · 2023-11-20
>
> Dear Reviewer C44C,
>
> Thank you for your detailed review and feedback. We have revised the manuscript according to your feedback, addressing W1, W2, W3, and W5.
>
> W1: See Figure 1 in new manuscript
>
> W2: See new Appendix C.1
>
> W3: See footnote 5 in Section 5.2
>
> W5: We have added sentence in the related work (highlighted in red) highlighting limitations of prior approaches and how we build upon them.
>
> Please take a look and let us know if these concerns have been adequately addressed. Please also let us know if there are any baselines you have in mind for W4, and we will try our best to address them in the limited available time we have left.
>
> Finally, as the discussion period is coming to a close, we would like to ask if there is anything else we could address. If not, we would appreciate if you could raise your score. Thanks again for your helpful feedback and review.

---

### Meta-Review · Area_Chair_TpW2 · 2023-12-11

**Metareview:**

**Summary**: The paper explores integrating language models (LMs) as probabilistic priors in non-linguistic tasks like semantic segmentation, household navigation, and activity recognition. The approach involves casting labeling and decision-making as inference in probabilistic graphical models, where LMs parameterize prior distributions. This method aims to incorporate uncertain observations and incomplete background knowledge effectively. The paper demonstrates how this approach improves predictions on rare, out-of-distribution, and structurally novel inputs across various tasks.

**Strengths**: The paper's main strength is its innovative use of LMs as priors for enhancing non-linguistic tasks, demonstrating versatility across different domains. The empirical studies, covering diverse objectives, show improvement across in-distribution, out-of-distribution, and zero-shot generalization scenarios. This suggests the method's effectiveness in domains beyond standard language processing tasks, offering a new perspective on integrating language understanding in various applications. Additionally, the approach is cost-effective compared to existing methods like Socratic Modeling.

**Weaknesses**: One significant weakness is the scalability and applicability focusing on tasks where LMs are expected to excel, without exploring cases where LMs may underperform. This limitation might hinder the approach's applicability in broader, more complex domains. The paper also lacks an in-depth exploration of LMs' potential negative impact on downstream tasks, such as scenarios involving complex spatial reasoning or unexpected object combinations. Moreover, the method appears to require substantial domain-specific information, which could limit its scalability and generalizability to wider applications.

Authors' Response to Weaknesses: The authors addressed concerns about the method's application scope and potential limitations. They emphasized the generality of LMs' priors and their alignment with real-world tasks, noting that the tasks were not specifically tailored for LMs. The authors also acknowledged the need to provide examples of LM failures and promised to include more qualitative examples in future versions. They discussed the guidelines for designing graphical models and prompts for LMs, highlighting the approach's flexibility and adaptability across different domains.

**Justification For Why Not Higher Score:**

This paper is on the borderline of being rejected. While the idea is novel and interesting, the experiment results do not support the claim enough. However, considering the novelty and interestingness of the paper, I would not object to accepting it.

**Justification For Why Not Lower Score:**

N/A

---

### Decision · Program_Chairs · 2024-01-16

Reject